# ADVERSARIALLY ROBUST NEURAL NETWORKS VIA OPTIMAL CONTROL: BRIDGING ROBUSTNESS WITH LYAPUNOV STABILITY

## ABSTRACT

Deep neural networks are known to be vulnerable to adversarial perturbations. In this paper, we bridge adversarial robustness of neural nets with Lyapunov stability of dynamical systems. From this viewpoint, training neural nets is equivalent to finding an optimal control of the discrete dynamical system, which allows one to utilize methods of successive approximations, an optimal control algorithm based on Pontryagin's maximum principle, to train neural nets. This decoupled training method allows us to add constraints to the optimization, which makes the deep model more robust. The constrained optimization problem can be formulated as a semi-definite programming problem and hence can be solved efficiently. Experiments show that our method effectively improves deep model's adversarial robustness.

## 1 INTRODUCTION

Deep neural networks achieve state-of-the-art performances on a variety of tasks (LeCun et al., 2015). However, neural nets are known to be vulnerable to adversarial examples. Imperceptibly perturbed inputs can induce erroneous outputs in neural nets (Szegedy et al., 2013). In image classification problems of computer vision, previous work has proposed various methods to attack deep models and induce low accuracy (Goodfellow et al., 2015; Madry et al., 2017; Papernot et al., 2016a; Carlini & Wagner, 2017a). Whereas multiple defenses against adversarial attacks are developed, they don't ensure safety faced with strong attacking methods. There are also theories that explain the existence of adversarial examples (Ilyas et al., 2019; Shamir et al., 2019), but they often fail to fully explain the features and behaviors of this phenomenon. This makes the study of adversarial attacks important in that it is a threat to real-life machine learning systems (Kurakin et al., 2016).

In this paper, we propose a dynamical system view on the adversarial robustness of the models, as well as new method that significantly defense adversarial attacks.

Recent works have shown the connection between deep neural networks and dynamical systems (E, 2017; Li et al., 2017; Haber & Ruthotto, 2017; Lu et al., 2017). If we regard the neural net as a discretization of an ordinary differential equation (ODE), then training neural nets becomes finding an optimal control of the corresponding discrete dynamical system. Traditionally, we often treat training neural networks as an unconstrained non-convex optimization problem

$$\min_{\theta \in \Theta} J(\theta) + R(\theta),$$

where $\theta$ denotes the parameters of the model, $J$ denotes the loss function and $R$ denotes the regularizer term, and we solve the problem with (stochastic) gradient-descent based methods (Bottou, 2010; Ruder, 2016). In the training process, we feed the network with a batch of training data, and compute the gradient with forward and backward propagation (E. Rumelhart et al., 1986). The propagation process resembles solving optimal control problems that tune the parameters to make the output be close to target states. This viewpoint motivates us to bridge adversarial robustness with Lyapunov stability of a dynamical system, and to train robust networks with algorithms that find stable optimal control. We will formulate the discussion in later sections.

## 2 RELATED WORK

### 2.1 ADVERSARIAL DEFENSE

Many defense methods have been proposed to improve the models' adversarial robustness. The defenses mainly fall into three types: adversarial training (Szegedy et al., 2013; Zhang et al., 2019), modifying the networks (Gu & Rigazio, 2015; Lyu et al., 2015; Papernot et al., 2016b; Nayebi & Ganguli, 2017; Ross & Doshi-Velez, 2017), and adding external models (Lee et al., 2017; Akhtar et al., 2017; Gebhart & Schrater, 2017; Xu et al., 2018; Sun et al., 2019). Although various defense methods have been developed, a defended deep model is often successfully attacked by newly developed attacks or specific counter-counter measures (Carlini & Wagner, 2017b). Therefore, it can be hoped that defenses against general attacks will be devised to make deep learning models (adversarially) robust to real-life threats.

### 2.2 NEURAL ODES AND OPTIMAL CONTROL

Recent works have bridged deep neural networks with ODEs and dynamical systems. On the one hand, deep residual networks (He et al., 2015) can be illustrated as forward Euler scheme approximating an ODE (E, 2017), which motivates us to design effective network structures (Lu et al., 2017). On the other hand, regarding the network as a dynamical system allows us to set up an optimal control viewpoint of neural nets. Pontryagin's Maximum Principle (Boltyanskii et al., 1960) has been applied to train neural nets (Li et al., 2017; Li & Hao, 2018).

## 3 ADVERSARIAL ROBUSTNESS AND LYAPUNOV STABILITY

### 3.1 DYNAMICS OF DEEP NEURAL NETS

Given a $T$-layer neural net, we let the dynamical system $\{f_t(x_t, \theta_t) : t = 0, \ldots, T\}$ represents the network, where $x_t$ is the input of $t$-th layer, $\theta_t$ is the parameter, and $f_t : \mathbb{R}^{d_t} \times \Theta_t \to \mathbb{R}^{d_{t+1}}$ denotes the $t$-th layer's transformation, which is usually a non-linear function $\sigma(\theta_t x_t + b_t)$ for fully-connected layers, convolution layers and batch normalization layers, etc. Therefore, training the neural net can be regarded as controlling the parameters to let the dynamics fit the training data. Specifically, the training optimization problem can be formulated as a typical optimal control problem as follows:

$$\min_\theta \sum_{i=1}^{B} J(x_T^i) + \sum_{i=0}^{T} L(\theta_i),$$

$$\text{subj. to } x_{t+1}^i = f_t(x_t^i, \theta_t), \ t = 0, \ldots, T-1,$$

where we use $x^i$ to denote the $i$-th input in the batch and $B$ denote the batch size. $J$ and $L$ are the loss function and the regularizer, respectively. Specially, if the model is a deep residual network with structure $x_{t+1} = x_t + f_t(x_t, \theta_t)$, we can regard the problem as the forward Euler discretization of the following continuous optimal control problem:

$$\min_\theta J(x(T)) + \int_0^T L(\theta(t)) \, \mathrm{d}t,$$

$$\text{subj. to } \dot{x} = f(t, x(t), \theta(t)), \ x(0) = x, \ 0 \le t \le T,$$

where $x(t)$ is a continuous trajectory from the input to the output logits.

### 3.2 LYAPUNOV STABILITY

Adversarial examples are usually clean images added by a small calculated perturbation $\eta$. The model predicts correct labels fed with clean inputs $x_0$, while the output is completely different when it is fed with perturbed input $x_0 + \eta$. The dynamical system view of neural nets motivate us to characterize this sensitivity with **Lyapunov stability** of a system (Hirsch et al., 2004).

**Definition 1** (Lyapunov Stability). *For a given dynamical system $\dot{x} = f(x), x(0) = x_0$, $x_e$ is an equilibrium, then*

- *The system is Lyapunov stable, if, $\forall \epsilon > 0$, $\exists \delta > 0$ such that, if $\|x(0) - x_e\| < \delta$, then for every $t \geq 0$, $\|x(t) - x_e\| < \epsilon$.*

- *The system is asymptotically stable if it is Lyapunov stable and $\exists \delta > 0$ such that if $\|x(0) - x_e\| < \delta$, then $\lim_{t \to \infty} \|x(t) - x_e\| = 0$.*

- *The system is exponentially stable if it is asymptotically stable and $\exists \alpha > 0, \beta > 0, \delta > 0$ such that if $\|x(0) - x_e\| < \delta$, then $\|x(t) - x_e\| \leq \alpha \|x(0) - x_e\| e^{-\beta t}$, for all $t \geq 0$.*

The definitions can be easily extended to discrete-time systems.

Intuitively, the Lyapunov stability states that for any small perturbation $\eta$, the trajectory is still "close enough" to the original one. If we regard a neural net as a dynamical system, and ensure the network is Lyapunov stable, then the model is robust to all (adversarial) perturbations.

### 3.3 ADVERSARIALLY ROBUST NEURAL NETS

Due to the connection between numerical ODEs and residual networks, we first consider robustness (i.e. Lyapunov stability) of continuous ODEs.

**Theorem 1** (Stable ODEs). *For a given ODE $\dot{x} = f(t, x, \theta) = \sigma(Ax + b)$, where $\sigma$ is the activation function, e.g., Sigmoid function or ReLU function, it is stable if $Re(\lambda_i(A)) \leq 0$, $\forall i$, where Re denotes the real part, and $\lambda_i$ denotes the $i$-th eigenvalue.*

One can see, e.g. Hirsch et al. (2004), for the proof of this theorem.

Theorem 1 provides a set of conditions for stable ODEs. However, deep residual network is only a forward Euler discretization scheme of continuous ODE. To ensure numerical stability, we require $|1 - \lambda_i(A)h| \leq 1$ (Ascher & Petzold, 1998), where the step size $h = 1$ in residual networks. Added by the identity mapping in residual networks, we can get the stable conditions for discrete dynamics.

**Theorem 2** (Stable Discrete Networks). *For a discrete neural network, i.e., discrete dynamics $\{f_t(x_t, \theta_t) : t = 0, \ldots, T\}$, where $f_t(x_t, \theta_t) = \sigma(\theta_t x_t)$ (we omit the bias term for simplicity), the network is stable if the $\rho(\theta_t) \leq 1$, where $\rho(A) = \max_i(|\lambda_i(A)|)$ is the spectral radius.*

If the conditions are added to the unconstrained optimization problem of training, we can greatly improve the adversarial robustness of neural nets. The methods will be discussed in the following section.

## 4 TRAINING ROBUST NEURAL NETS

### 4.1 PMP AND MSA

For deterministic systems, the Pontryagin's Maximum Principle (PMP) (Boltyanskii et al., 1960) provides a set of necessary conditions for optimal control of the system. Various algorithms have been proposed to solve the deterministic optimal control problem based on PMP. Among them, the Method of Successive Approximations (MSA) (Krylov & Chernous'ko, 1963) is one of the simplest algorithms. In the field of deep learning, previous work has utilized MSA to train neural networks (Li et al., 2017; Li & Hao, 2018).

Formally, consider the optimal control problem for training neural nets in section 3. For dynamics $\{f_t(x_t, \theta_t) : t = 0, \ldots, T\}$, assume $\theta^* = \{\theta_0^*, \ldots, \theta_{T-1}^*\}$ is a solution to the optimal control problem. Also, we define the Hamiltonian function $H : \mathbb{R}^{d_t} \times \mathbb{R}^{d_{t+1}} \times \Theta_t \times [T] \to \mathbb{R}$ by $H(x, p, \theta, t) = p \cdot f_t(x, \theta) - L(\theta_t)$, where the dot denotes the inner product. We have the following necessary conditions for $\theta^*$.

**Theorem 3** (Pontryagin's Maximum Principle for Discrete Systems). *Assume $f_t$ and $J$ are sufficiently smooth. There exists co-states $p^* = \{p_0^*, \ldots, p_T^*\}$ s.t. the following conditions hold:*

$$x_{t+1}^* = \nabla_p H(x_t^*, p_{t+1}^*, \theta_t^*, t),\ x_0^* = x_0,$$
$$p_t^* = \nabla_x H(x_t^*, p_{t+1}^*, \theta_t^*, t),\ p_T^* = -\nabla_x J(x_T^*),$$
$$\theta_t^* = \arg\max_{\theta} H(x_t^*, p_{t+1}^*, \theta, t).$$

For simplicity of notations, here we assume the batch size is 1. One can easily extend the theorem to minibatch training case by summing over the batch.

The theorem can be proved by KKT conditions (Boyd & Vandenberghe, 2004), where the co-states can be seen as the Lagrangian dual variables.

Consider the conditions in PMP, one can find the $x$ equations are exactly the forward propagation of a neural net, and the $p$ equations resemble the backward propagation process. The third condition states that the model parameters must maximize the Hamiltonian function. This motivates us to iteratively compute forward and backward propagation, and solve the Hamiltonian maximization to find the optimal control, which is exactly the Method of Successive Approximations (Algorithm 1). In practice, we usually add regularizer terms that penalize great changes in the maximization step to prevent drastic steps that cause divergence. For the connection between MSA and back-propagation-based gradient descent algorithms, see the appendix of Li & Hao (2018).

---

**Algorithm 1** The Method of Successive Approximations
___

Initialize $\theta^0 = \{\theta_0^0, \ldots, \theta_{T-1}^0\}$, set $k = 0$;
**repeat**
    Compute the states (forward propagation): $x_{t+1} = \nabla_p H(x_t, p_{t+1}, \theta_t^k, t)$, $t = 0, \ldots, T-1$;
    Compute the co-states (backward propagation): $p_t = \nabla_x H(x_t, p_{t+1}, \theta_t^k, t)$, $t = T-1, \ldots, 0$, with initial $p_T = -\nabla_x J(x_T)$;
    For each $t = 0, \ldots, T-1$, solve the maximization $\theta_t^{k+1} = \arg\max_\theta H(x_t, p_{t+1}, \theta, t)$;
    Set $k = k + 1$;
**until** Converge;

---

The advantages of training by MSA compared with gradient descent algorithms has been discussed in (Li et al., 2017), among which the most significant feature is that the optimization steps on different layers are decoupled. Concretely, after computing the states $x$ and co-states $p$, the optimization step on layer $t$ is only searching for parameters $\theta_t$. This not only suggests that the optimization process can be accelerated by parallelization, but also allows us to utilize the features of the problem. The parameter space is greatly reduced compared with the original intractable optimization problem, and hence the optimization is much more easier. This allows us to add constraints that ensure robustness of the model.

## 4.2 ROBUST CONSTRAINTS

Consider a layer in the form of $f_t(x) = \theta_t x$, where we leave the activation as an individual layer with no parameters for simplicity, we can derive the following optimization problem for Hamiltonian maximization:

$$\max_\theta p_{t+1} \cdot (\theta_t x_t) - \alpha \|\theta_t\|_2^2 - \beta \|\theta_t - \theta_t'\|_2^2,$$

$$\text{subj. to } \rho(\theta_t) \leq 1,$$

where $\alpha \|\theta_t\|_2^2$ is the L$_2$ norm regularizer (weight decay), and $\theta_t'$ is the initial parameter (i.e., $\theta_t^k$ in the algorithm). The last term keeps the training process from drastic steps that cause divergence. The constraint, as illustrated in section 3, is the stable condition for discrete systems. It makes the optimization quite difficult if we directly add the constraints in gradient descent based algorithms, but the decoupled optimization in MSA allows us to do so.

With regard to the constraint of parameter's spectral radius, a simple method is to apply special forms of matrices for parameters, e.g. anti-symmetric matrices. For continuous deep models, the only constraint is Theorem 1, i.e., $\text{Re}(\lambda_i(\theta_t)) \leq 0$. Anti-symmetric matrices have only imaginary eigenvalues, and hence we can replace $\theta_t$ with $\theta_t - \theta_t^T - \gamma I$, where $\gamma$ is a small positive constant.

For general forms of parameters, one can prove the following transformation.

**Theorem 4.** *One sufficient condition of $\rho(A) \leq 1$ is*

$$\begin{bmatrix} I & A \\ A^T & I \end{bmatrix} \succeq 0,$$

*where $A \succeq B$ denotes $A - B$ is positive semi-definite.*

Table 1: Results of robust training on CIFAR10.

| Method | Vanilla | Adv Training | Ours |
|---|---|---|---|
| Clean data | 80.73% | 79.61% | 74.49% |
| FGSM (Goodfellow et al., 2015) | 2.34% | 77.45% | 49.32% |
| PGD-10 (Madry et al., 2017) | 0.02% | 46.67% | 36.33% |
| C&W (Carlini & Wagner, 2017a) | 0.01% | 21.15% | 16.80% |

*Proof.* Recall that $\rho(A) \leq \|A\|_2 = \sqrt{\lambda_{\max}(A^T A)}$, we have

$$\|A\|_2 \leq 1 \Leftrightarrow A^T A \preceq I \Leftrightarrow \begin{bmatrix} I & A \\ A^T & I \end{bmatrix} \succeq 0.$$

$\square$

Hence we can replace $\rho(\theta_t) \leq 1$ with a positive semi-definite condition, and we turn the Hamiltonian maximization into a new optimization problem, where the target function is quadratic and the constraint is a semi-definite condition. This can be reduced to a semi-definite programming (SDP) problem (Vandenberghe & Boyd, 1998), which is a special case of convex optimization, and thus can be solved efficiently by, e.g., interior point methods (Helmberg et al., 1970) in polynomial time.

Here we summarize our method. For a given neural network, we use MSA to train the model, i.e., iteratively computing the states (forward propagation) and co-states (backward propagation), and solving the optimization for each layer. Instead of directly maximizing the Hamiltonian, we add a positive semi-definite constraint to the optimization problem, which leads to a stable control of the dynamics.

## 5 EXPERIMENTS

### 5.1 EXPERIMENT SETUP

To evaluate the effectiveness of our method, we conduct experiments on CIFAR10. We trained the network on clean data, with adversarial training (PGD-10) and with robust training (our method), respectively. We used FGSM (Goodfellow et al., 2015), PGD-10 (Madry et al., 2017) and C&W (Carlini & Wagner, 2017a) to attack the network.

Due to the limitation of TensorFlow, we used a simple interior point method with gradient descent to solve SDP. The network model was an 18-layer residual network (He et al., 2015), with 8 residual blocks. We set the perturbation size as $\epsilon = 0.1$ for both FGSM and PGD. For C&W, we used the $L_0$ metric. We trained the model for 150 epochs with a batch size of 200. The learning rate was set to be $10^{-2}$ initially, and was divided by 5 at epoch 30, 60 and 100. The regularizer term constant was set to be $10^{-3}$.

### 5.2 RESULTS

The results can be seen in Table 1. The accuracy of robust models on clean data is lower than vanilla model's in that robust training and generalization is more difficult and requires more data (Schmidt et al., 2018).

Our method improves model's adversarial robustness, compared with the vanilla model. Figure 1 displays the eigenvalues of the last fully-connected layer's parameter. The complex norm of eigenvalues (spectral radius) of the model trained by our method are effectively bounded below 1, which satisfies the robust constraint on parameters in section 4.2, while eigenvalues of natural training are randomly distributed in the complex plane.

Our method is not as effective as traditional adversarial training method. However, it mainly has the following advantages: (a) The training process doesn't require large numbers of gradient propagation, which consumes much time in adversarial training. In our experiment, adversarial training

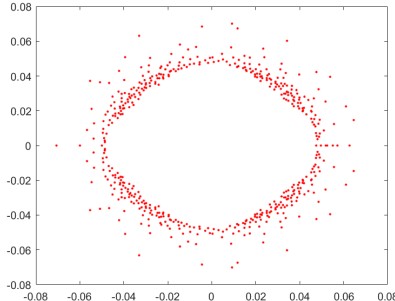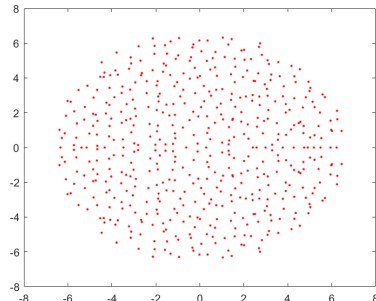

Figure 1: Eigenvalues of the last fully-connected layer for robust training (left) and vanilla training (right).

spends about 10 times GPU time as much as our method. (b) The decoupled training process allows us to set different hyperparameters and training methods for different layers, which is more maneuverable for large scale training. We can further control the behavior of different layers in adversarial settings. (c) Lyapunov stability provides a framework for analyzing adversarial robustness of deep models, which may lead to theoretical analysis of adversarial samples in future work.

## 6 DISCUSSION AND FUTURE WORK

Motivated by the dynamical system view of neural networks, this work bridges adversarial robustness of deep neural models with Lyapunov stability of dynamical systems, and we also propose a method that uses a stable optimal control algorithm to train neural networks to improve the adversarial robustness of deep neural models. Though the result didn't surpass STOA defense methods, the stable control view of training neural nets points out another direction towards adversarially robust models.

For future work, on the one hand, mathematical analysis on Lyapunov stability of neural models may be studied to provide theoretical understanding of adversarial robustness. On the other hand, popular platforms for deep learning, e.g., TensorFlow, PyTorch, didn't provide frameworks for optimal control. We will obtain better results if specific algorithms for SDP are applied to solve the optimization problem.

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
