# OpenReview forum: "Adversarially Robust Neural Networks via Optimal Control: Bridging Robustness with Lyapunov Stability"
_ICLR.cc/2020/Conference — Reject_

### Official Review · AnonReviewer3 · 2019-10-22
**Official Blind Review #3**

**Rating:** 1

**Review:**

Summary:
The goal of this paper is to train neural networks (NNs) in a way to be robust to adversarial attacks. The authors formulate training a NN as finding an optimal controller for a discrete dynamical system. This formulation allows them to use an optimal control algorithm, called method of successive approximations (MSA), to train a NN. The authors then show how constraints can be added to this optimization problem in order to make the trained NN more robust. They show that the resulted constraint optimization problem can be formulated as a semi-definite programming and provide some experimental results.

Comments:
- Although the problem studied in the paper is important and the approach is interesting, it seems the paper has been written in rush and in my opinion is not ready for publication. The writing is not good. The introduction and related work sections are incomplete and not very informative. It is not clear what has been done before and what is the contribution of this paper. The main technique/algorithm of the paper has not been explained clearly that someone can easily understand and implement it. The experimental results are not convincing.
- There are strong claims in the paper such as "experiments show that our method effectively improves deep model's adversarial robustness", this is too strong, given the quality of the experiments of the paper. Or "the constraint optimization problem can be formulated as a semi-definite programming (SDP) problem and hence can be solved efficiently", to the best of my knowledge, SDP solvers are limited to small problems and cannot solve the large problems efficiently.
- The area of making NNs robust to attacks is a very active area and there are many attacks and solutions out there, which require more comprehensive empirical studies of any new method. I do not see this in the paper.
- Overall, I think this paper requires a major revision in order to be evaluated better and to be more useful for the community.

**Experience Assessment:**

I have read many papers in this area.

**Review Assessment: Checking Correctness Of Derivations And Theory:**

I assessed the sensibility of the derivations and theory.

**Review Assessment: Checking Correctness Of Experiments:**

I assessed the sensibility of the experiments.

**Review Assessment: Thoroughness In Paper Reading:**

I read the paper at least twice and used my best judgement in assessing the paper.

---

### Official Review · AnonReviewer1 · 2019-10-23
**Official Blind Review #1**

**Rating:** 6

**Review:**

The paper contributes to the robust training of neural networks as follows:
  1) The paper uses the theoretical view of a neural network as a discretized ODE to develop a robust control theory aimed at training the network while enforcing robustness;
  2) Such an objective is achieved by introducing Lyaponov stability and practically implemented through the method of successive approximations;
  3) Empirical evaluation demonstrate that the newly introduced method performs as well as the SOTA in terms of defensive training.

The paper is well written and proposes a well motivated and theoretically original strategy to robustly train neural networks against adversarial examples.
The strength of the paper is definitively in its theoretical section, it would be really great to see an empirical improvement improvement on the SOTA.
However, I do not believe the paper should be penalized for only matching other algorithm as it relies on a tractable and principled theoretical analysis.

**Experience Assessment:**

I have read many papers in this area.

**Review Assessment: Checking Correctness Of Derivations And Theory:**

I assessed the sensibility of the derivations and theory.

**Review Assessment: Checking Correctness Of Experiments:**

I assessed the sensibility of the experiments.

**Review Assessment: Thoroughness In Paper Reading:**

I read the paper at least twice and used my best judgement in assessing the paper.

---

### Official Review · AnonReviewer2 · 2019-10-24
**Official Blind Review #2**

**Rating:** 1

**Review:**

Neural Networks are vulnerable to adversarial perturbations. This paper proposes a method that based on optimal control theory that uses semidefinite-programming. This is a quite popular topic in Adversarial training recently, there has been a few works in that line. There are almost no experiments in this paper. There are several typos in the paper and writing of this paper requires more work. There are several typos in this paper, for example STOA, should be SOTA (in the Section 6.) In its current state, this paper looks very rushed.


As Yiping Lu pointed out, the PMP statement in this paper is also wrong. At this current stage, unfortunately this paper doesn’t meet the standards of ICLR. I would recommend the authors to go over the paper carefully and resubmit to a different venue.




**Experience Assessment:**

I have read many papers in this area.

**Review Assessment: Checking Correctness Of Derivations And Theory:**

I assessed the sensibility of the derivations and theory.

**Review Assessment: Checking Correctness Of Experiments:**

I assessed the sensibility of the experiments.

**Review Assessment: Thoroughness In Paper Reading:**

I made a quick assessment of this paper.

---

### Public Comment · ~Yiping_Lu1 · 2019-10-03
**PMP has already appears in YOPO and this version theorem is wrong**

Dear authors ,
I'm one of the authors of the paper "You only propagate once".  First, thanks for the citation, but we still have some points need to be point out.
- YOPO is also a method using PMP, you can see our paper section 3 in our paper for details.
- The PMP you state in the paper is wrong, for using KKT condition you can only have *\nabla H=0* to get \theta = argmax H, you need more assumption {ft(x, θ) : θ ∈ Θt} is convex.(notation in our paper)
- I also curious about the training speed about your method compare with YOPO. Whether there is some code for us to try?

---

> ### Author Response · Authors · 2019-10-04
> **Thanks for your comment!**
>
> Hi,
>
> Thanks for your comment!
>
> - YOPO is a great work on PMP and it inspires a lot.
> - Thanks for pointing out the mistake. I didn't scrutinize the theorem so that I missed the convex constraint and I will fix it.
> - Actually, the code is not well-organized. I'm still working on it and will release it upon finishing.

---

> > ### Public Comment · ~Yiping_Lu1 · 2019-10-05
> > **Additional Comment**
> >
> > Thus what is the difference between PMP here  and PMP in YOPO.
> > and
> > what is the benefit of using  Successive Approximations in adversarial training?
> > (at least you should have a speed comparison to demonstrate the benefit
> >
> > You said "The advantages of training by MSA compared with gradient descent algorithms has been discussed
> > in (Li et al., 2017)" they said their benefit is avoiding saddle point
> > Why  Successive Approximations can avoid saddle point in the adversarial training? (first what is "saddle point" in min-max optimization?)

---

> > > ### Author Response · Authors · 2019-10-05
> > > **Answers to the question**
> > >
> > > I'm not sure I understand your question. Our method is not a modification of adversarial training. The reason why we use MSA is to reduce the size of the parameter space to add extra constraints.

---

### Decision · Program_Chairs · 2019-12-19

**Decision:**

Reject

**Comment:**

The authors propose a framework for improving the robustness of neural networks to adversarial perturbations via optimal control techniques (Lyapunov Stability and the Pontryagin Maximum Principle, in particular). By considering a continuous-time limit of the training process, the authors use the PMP to derive udpate rules for the neural network weights that would result in a robust network. While the approach is interesting, the paper has some serious deficiencies that make it unacceptable for publication in its current form:

1. Quality of empirical evaluation: The authors only report final numbers on CIFAR-10 for a fixed set of adversarial attacks. It has been observed repeatedly in the adversarial robustness literature that adversarial evaluation of neural networks has to be done carefully to guard against possible underestimation of the worst-case attack. In particular, the specific details of the adversarial attacks used (number of steps, number of initializations, performance under larger perturbation radii) that are necessary to trust the results are not given (see https://arxiv.org/pdf/1902.06705.pdf for example).

2. Unclear novelty: The authors do not sufficiently explain the novelty in their approach relative to prior work (particular prior work that has used optimal control ideas in this context).

3. Computational cost: The authors do not give sufficient details to judge the computational overhead of their method to judge how much more expensive it is to train with their approach relative to standard or adversarial training.

While one reviewer voted for a weak accept, the other reviewers were in consensus on rejection. The authors did not respond during the rebuttal phase and hence the reviews were unchanged.

In summary, I vote for rejection. However, I think this paper has potentially interesting ideas that should be carefully developed and evaluated in a future revision.